# Make Time for Employees to Be Sustainable: The Roles of Temporal Leadership, Employee Procrastination, and Organizational Time Norms

**Juncheng Zhang *** , **Shuyu Zhang, Fang Liu and Weiqi Chen ***

School of Management, Guangzhou University, Guangzhou 510006, China;
zhangshuyu@rongxin-consulting.com (S.Z.); liufang@gzhu.edu.cn (F.L.)
* Correspondence: zhangjuncheng@gzhu.edu.cn (J.Z.); paulchen8903@163.com (W.C.)

**Abstract:** Extended work availability (EWA) captures the experience of an employee who needs to be available for job demands during nonworking hours. It is a ubiquitous phenomenon because of the prevalent use of information and communication technology (ICT) such as mobile devices and internet services for work purposes. Although it has been found to impair employee health and well-being, evidence that delineates how to mitigate employee EWA is sparse. Thus, an important research question is: How can managers alleviate employee EWA in the ICT-prevalent work environment? Given EWA has a close connection with the time-based work–nonwork conflict, the present study addresses this question by taking a temporal lens and focusing on the roles of three time-related determinants of employee EWA. Particularly, we first include temporal leadership as a predictor of employee EWA, which concerns a particular type of time management behavior in which a manager aims at helping employees to achieve effective use of time while performing job duties. Then, we incorporate both the individual tendency to delay an intended course of action (i.e., procrastination) and the time management environment in an organization (i.e., organizational time norms) into our research model to further reveal how employee EWA could be shaped. Drawing on spillover theory, the goal of the present study was to examine the effect of temporal leadership in determining employee EWA, as well as the roles employee procrastination and organizational time norms play. Analyses of cross-sectional survey data from a sample of 240 full-time employees showed that temporal leadership has a U-shaped association ($\beta = 0.32$, $p < 0.001$) with employee EWA. Both employee procrastination ($r = 0.40$, $p < 0.001$) and organizational time norms ($r = 0.30$, $p < 0.001$) are positively related to employee EWA, respectively. Moreover, the U-shaped association between temporal leadership and employee EWA becomes more salient when the organizational time norms is strong, with a standardized regression coefficient of 0.24 ($p < 0.05$) for the interaction between temporal leadership squared and organizational time norms. These findings contribute to a more comprehensive view of how managers can alleviate employee EWA in today's ICT-prevalent work environment.

**Keywords:** extended work availability; temporal leadership; procrastination; organizational time norms; spillover theory

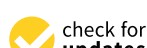



## 1. Introduction

In the past few decades, a variety of information and communication technologies (ICT) have been adopted in the workplace such that reshape the world of work to a great extent. For example, using mobile devices and internet services enables employees to perform job duties anytime and anywhere. Although ICT has the capacity of enabling employees to handle work matters more efficiently [1] and to reconcile their personal and work lives [2], employees who work outside the employer's premises via ICT usually work longer [3]. The reason for this is usually associated with a phenomenon that has increasingly

drawn scholarly attention [4,5], namely, extended work availability (EWA). EWA refers to "a condition in which employees formally have off-job time but are flexibly accessible to supervisors, coworkers, or customers and are explicitly or implicitly required to respond to work requests (p. 106)" [6]. Prior research has shown that EWA is associated with employee health problems, stress perceptions, and low recovery levels [7–9], which might further inhibit employees from sustainably making contributions to their organization with a high level of thriving at work (vitality and learning) [10,11]. These consequences are related to the health dimension of human capital [12], and the latter is one of the critical determinants of organizations' sustainability [13]. Given managers often play a critical role in determining employees' working and nonworking lives [14,15], the present study strives to answer the following research question: How can managers alleviate employee EWA in the ICT-prevalent work environment?

Valuable knowledge on this topic has emerged from at least two types of study. First, technology-oriented research usually stresses the role of the latest technologies adopted in the workplace. For example, artificial-intelligence-supported workplace decisions or algorithmic decision making in organizations usually have the capacity to increase productivity in a shorter time with less manpower. By facilitating the adoption of these technologies in the workplace, managers might help employees to decrease their working time, especially beyond normal business hours. Second, management-oriented research tends to emphasize the role of managerial strategies managers take in the workplace. For example, Matusik and Mickel found that supervisors' expectations about responding quickly obliges employees to be accessible and responsive after normal business hours [16]. In addition, Cavazotte et al. found that the constant connectivity demands from supervisors invaded employees' life domain and negatively affected their private sphere [17]. In a recent narrative synthesis of prior research, Schlachter et al. identified expectations, demands, and supports from one's supervisor as the essential elements for the organizational context in predicting employee extended work hours [18]. Compared with technology-oriented research, management-oriented research lays a more promising direction towards answering our research question.

However, significant theoretical gaps persist on this topic. First, although great efforts have been devoted to investigating the negative effects of EWA on employees' health problems and well-being [6,7], far less attention has been paid to the determinants of employee EWA. This undermines managers' effectiveness in lessening employees' experience of EWA in today's ICT-prevalent work environment. Second, unlike several fragmented ICT-related managerial strategies that have been mentioned in extant research [16,17], knowledge is limited concerning how more specific leader behavior variables are associated with employee EWA. This hinders the theorization of the underlying mechanisms and boundary conditions of how managers can alleviate employee EWA. Third, although EWA requires individuals to meet work demands during off-hours, the temporal lens [19] has not been adopted to investigate why and how employees experience EWA. This is surprising, given that time is an essential facet in understanding how employees engage in home-based work as well as its consequences [20]. It has been argued that devoting nonworking time, especially leisure time, elicits individuals' time-based work–nonwork conflict [21] and hamper one's recovery from work [22,23]. Additionally, this may further jeopardize organizational sustainability [13].

To address these issues, the current study aims to advance knowledge on the emergence of employee EWA by incorporating temporal leadership as an antecedent. The discussion of workplace leadership usually includes a variety of influence processes that occur in organizations to warrant the achievement of common goals [24]. Unlike other established broad leadership theories (e.g., transformational leadership, participative leadership, adaptive leadership, etc.), temporal leadership follows the trends of other emerging theories that concern more specific leadership topics (e.g., ethical leadership, abusive supervision, and E-leadership) [25–27]. It concerns a particular type of time management behavior in which a manager aims at helping employees to achieve effective use of time

while performing job duties [28,29]. Given time serves as an important factor in understanding the work–home conflict related to EWA [20,21], temporal leadership should be especially relevant for orchestrating employees to effectively allocate time among work and nonwork activities. Particularly, in answering Pierce and Aguinis's call for studies on the too-much-of-a-good-thing (TMGT) effect in management [30], we examine the curvilinear relationship between temporal leadership and employee EWA.

We also introduce employee procrastination and organizational time norms in our conceptual model (see Figure 1). Procrastination refers to the individual tendency to delay an intended course of action [31]. Having a lack of self-control, procrastinators usually cannot complete tasks within regular business hours. Then, they might need to deal with work matters even in nonworking hours (i.e., EWA) to catch up with deadlines, especially without certain action-facilitating stimuli such as time pressure and positive affect [32]. In addition, procrastination could also play a role in shaping the effects of temporal leadership on employee EWA. Because temporal leadership might elicit employees' positive affect (e.g., vigor [33]) if it is moderately strong, this helps employees overcome procrastination and complete one's job within working hours [32]. Otherwise, an extremely strong temporal leadership might hamper employees' positive affect (e.g., job satisfaction [34]) because of the deprivation of autonomy in deciding how to use time themselves. Additionally, this inhibits employees from initiating tasks earlier, such that they need to handle unfinished work matters during off-time (i.e., EWA). Compared with procrastination, organizational time norms delineates the intangible and shared patterns of expected temporal activity at the organization level [19,35]. Previous research has shown that certain patterns regarding the usage of time (e.g., polychronicity [36]) would increase procrastinating behavior. We contend that organizational time norms might influence employee EWA because procrastinators usually must deal with work even outside the working hours if they cannot complete their job at work. Furthermore, organizational time norms serves as the time-related context in which temporal leadership is exerted. It could also interact with temporal leadership when influencing employee EWA. This is because previous research has found that the effects of temporal leadership depend on whether time is limited [37] or shared temporal cognitions exists [38] to some extent.

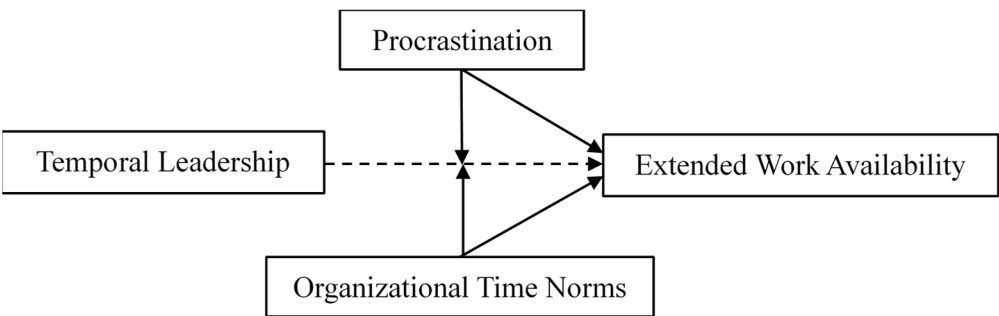

**Figure 1.** Conceptual model of the curvilinear relationship between temporal leadership and extended work availability. Note: The dotted arrow represents a U-shaped relationship.

This research contributes to the existing literature in three ways. First, it proposes a promising framework for exploring the determinants of employee EWA by taking the temporal lens [19] in organizational research. Focusing on the effect of temporal leadership, as well as the roles of employee procrastination and organizational time norms, this study is the first one in this field except for several fragmented arguments related to this topic [18,39]. Second, it advances the spillover literature to some extent by examining the curvilinear relationship between temporal leadership and employee EWA. Although certain work conditions (e.g., leadership practices) were argued to have both positive and negative direct effects on nonwork domain outcomes [21,40], prior studies tend to examine either positive or negative spillover effects separately. This study helps to extend the theoretical framework of spillover theory by integrating these two types of spillover effects into a same

model simultaneously. Third, it advances the emerging temporal leadership theory through providing some interesting evidence on its effects. Although temporal leadership has been widely accepted as a favorable leadership variable that has desired outcomes [29,33,41,42], its potential dark side seems to have been neglected. This study not only sheds new light on the effects of temporal leadership but also contributes to answering Pierce and Aguinis's call for studies on the TMGT effect in management [30].

## 2. Theoretical Background and Hypothesis Development

### 2.1. Temporal Leadership and Employee EWA

Temporal leadership denotes a set of time management behaviors undertaken by a leader to help employees achieve an effective use of time while performing job duties [28,29]. It captures the task-oriented behaviors of a leader that highlight temporality, such as helping employees schedule, synchronize, and allocate temporal resources at work [29,43]. These behaviors help employees to deal with a variety of time-related challenges, such as temporal ambiguity, conflicting temporal interests and requirements, and scarcity of temporal resources [44,45]. Since today's turbulent business environment needs leaders to become much more sensitive to the temporal needs in and of their organizations, integrating time-related practices into leadership research is instructive [19,46,47].

As the widespread use of ICT is increasingly blurring the boundary of work and non-work domains, and often extends employees' total working hours [3], temporal leadership might play a role in responding to these issues. By helping employees schedule tasks, synchronize paces, and allocate time at work [29,43], leaders can orchestrate employees to meet deadlines in an organized way. This can mitigate employees' suffering from time famine, which denotes "a feeling of having too much to do and not enough time to do it (p. 57)" [48]. Thus, temporal leadership is a "good thing (p. 121)" [49] and can serve, like other supervisor behaviors (e.g., family-supportive supervisor behavior [50]), as a job resource for employees. Since job resources have been related to a positive work–home interference [51,52], we expect temporal leadership to have a similar effect.

Drawing on spillover theory and relevant research [21,40,53], work and family have effects on one another such that generate similarities between the two domains. Except for the similarity between a work construct and a distinct but connecting family construct (e.g., job and life satisfaction [54]), spillover can also be characterized as experiences transferred intact between work and family domains (e.g., work fatigue displayed at home [55]). Focusing on the transferred or extended experiences, the second version of spillover does not involve a linkage between a work construct and a family construct [21]. With the help of a moderately strong temporal leadership in the workplace, individuals are less vulnerable to time pressure in meeting deadlines [42]. Specifically, employees might have a weaker feeling of being stuck at work. That is, it is not necessary for them to be available for the work demands without any break during business hours. Given that the essence of spillover can be captured by a process, namely extension [53], we expect such a relaxed experience induced by temporal leadership would spill over into one's nonwork domain. That means employees might be less likely to suffer from being available for their work demands outside regular working hours. As extended work availability portrays an employee's requirements to deal with work during off-time [6], we therefore expect a moderately strong temporal leadership would have a negative association with it.

An extremely strong temporal leadership, however, might have a TMGT effect [30]. This is because it usually captures the task-oriented rather than relationship-oriented behaviors of a leader that highlight temporality [29]. Orchestrating employees to meet time-related challenges at work often needs a leader to closely monitor their work pace or supervise their time-use behaviors. In a sense, leaders who display extremely strong temporal leadership tend to micromanage [56,57] their subordinates regarding how to meet deadlines. This is likely to undermine employees' autonomy in deciding how to schedule, synchronize, and allocate time at work. As such, employees should always stay ready to respond to the requests of their leaders in terms of meeting a variety of deadlines

and/or milestones at work. In line with the second version of spillover [21,53], such an experience of employees in the work domain might transfer or extend to the nonwork domain. Therefore, we expect that an extremely strong temporal leadership has a positive association with employees' feelings of being accessible to the work demands during off-time (i.e., EWA).

In sum, we contend that temporal leadership probably has a TMGT effect [30] on employee EWA. Specifically, a moderately strong temporal leadership alleviates employee EWA, while an extremely strong temporal leadership exacerbates employee EWA. This leads to the following hypothesis:

**Hypothesis 1.** *Temporal leadership has a U-shaped association with employee EWA.*

*2.2. The Effect of Employee Procrastination*

Procrastination refers to an individual tendency to "voluntarily delay an intended course of action despite expecting to be worse off for the delay (p. 66)" [31]. It is usually considered as a pernicious form of self-regulation failure [31,58]. Given its temporal property in delaying a planned course of action, procrastination often has a close association with the lack of time management skills [59,60]. Thus, individuals who are high in procrastination tend to have difficulty in completing their assigned tasks within the formal working time.

For those who can deal with work during off-time beyond normal workplaces, procrastination usually puts them under pressure to handle work matters even when they are off duty. In fact, procrastinators are likely to lose work–life balance under such flexible job conditions [61]. A recent study found that there is a positive association between procrastination and a prolonged working day [62]. Moreover, procrastination has also been found to have a negative association with employees' psychological detachment from work [63]. Therefore, employees with high procrastination are more likely to experience a heavy requirement to remain available for job demands on workdays. Additionally, in line with the second version of spillover [21,53], a similar experience might emerge in the nonwork domain. Consequently, employees who are high in procrastination probably suffer from the experience of EWA during off-time. This leads to the following hypothesis:

**Hypothesis 2.** *Procrastination has a positive association with employee EWA.*

Drawing on spillover theory, time allocation decisions may moderate the relationship between work time and family time [21]. As procrastination usually entails making decisions on how to allocate one's time into his/her planned and other actions [64,65], it might play a similar role when exploring the relationship between temporal leadership and employee EWA. Specifically, both the negative and positive associations would be enhanced for employees with high procrastination. That is because procrastination is a form of self-regulation failure [31,58]; employees with high procrastination usually have difficulties in managing their time effectively.

For employees with high procrastination, we argue that a moderately strong temporal leadership enables them to make better use of time. With leaders helping them schedule, synchronize, and allocate temporal resources [29,43], employees with high procrastination can more easily achieve an effective use of time while performing job duties. Not to mention moderately strong temporal leadership might elicit employees' positive affect (e.g., vigor [33]), which helps procrastinating employees overcome procrastination [32]. As such, they are less likely to be pressed to meet deadlines [42] again and again within regular hours. In a sense, they might be somewhat relieved from the nonstop availability of the job demands, especially when deadlines are approaching. Following the experience spillover process [21,53], employees with high procrastination will be able to alleviate their suffering of EWA with the help of a moderately strong temporal leadership.

When coupled with an extremely strong temporal leadership, however, high procrastination could exacerbate the pernicious effect of temporal leadership on employee

EWA. On the one hand, managers who display extremely strong temporal leadership deprive the time-related autonomy of employees regarding how to meet deadlines through micromanagement [56,57]. This might hamper employees' positive affect (e.g., job satisfaction [34]) because of the deprivation of autonomy in deciding how to use time themselves. In this vein, an extremely strong temporal leadership would inhibit employees from initiating tasks earlier, such that they need to handle unfinished work matters during off-time (i.e., EWA). On the other hand, employees with high procrastination usually voluntarily delay the planned course of action [31] until they truly decide to accomplish it. Thus, conflicts regarding how to use time are likely to occur when procrastinating employees work under an extremely strong temporal leadership. This kind of inconsistency has been argued to hinder leader–follower coordination [66]. In addition, procrastination has been found to have a positive association with lack of control [67]. It can be expected that procrastinators, who fail to regulate themselves [24,68], are pushed further by an extremely strong temporal leadership to keep up with work time and time again. Moreover, procrastinators have also been found to fail in leading a high quality of life [69]. Therefore, we draw on spillover theory [21,53] and contend that employees' intensified suffering of being available for job demands transfers into their nonwork domains (i.e., EWA).

Based on the arguments above, we argue that procrastination serves as a moderator in the curvilinear association between temporal leadership and employee EWA. For employees with high procrastination, a moderately strong temporal leadership could be more inclined to mitigate employee EWA, while an extremely strong temporal leadership could be more inclined to aggravate employee EWA. This leads to the following hypothesis:

**Hypothesis 3.** *Procrastination moderates the U-shaped association between temporal leadership and employee EWA, such that both the negative and positive associations are enhanced when employees are high in procrastination.*

### 2.3. The Effect of Organizational Time Norms

Organizational time norms denote the intangible and shared patterns of expected temporal activity at the organization level [19,35]. It can be considered as an organization's time management environment which has effects on how individuals use time at work [35]. Specifically, this environment can be reflected by several dimensions related to supervision, coworker interaction, job descriptive processes, support for time management processes, and time values [70]. If the time management practices are facilitated by managers, coworkers, and the process and/or policy in organizations, employees are probably shaped to stress time-related issues (e.g., deadlines, punctuality, work speed, timing, scheduling, prioritizing, etc.) while performing duties.

Organizational time norms have a necessary connection with time-related moral issues [71], they influence employees' time-use behaviors mainly through social pressure [35]. Specifically, emphasizing the value and effective use of time by organizations usually obliges employees to follow organizational time norms. By breaching the organizational time norms, however, employees may be confronted with the loss of reputation, being ostracized, or even laid off because of evoking intense reactions from peers [35]. For example, if time is viewed as an important resource in organizations, employees who usually perform job duties without prioritizing or fail to meet deadlines will be complained about by colleagues that collaborate with them. To avoid this undesirable consequence, employees may keep themselves working on their job duties during working hours. Drawing on the experience spillover process [21,53], we expect the availability of employees at work "extends" to the nonwork domain (i.e., EWA). Thus, this leads to the following hypothesis:

**Hypothesis 4.** *Strong organizational time norms have a positive association with employee EWA.*

Since organizational time norms might constrain employees' time management behaviors through social pressures [35], they undermine the freedom of employees' time

allocation decisions. Given that time allocation decisions can moderate the relationship between work time and family time [21], we herein propose organizational time norms can also play a moderating role in the relationship between temporal leadership and employee extended work availability. Specifically, strong organizational time norms may heighten both the negative and positive associations between these two constructs. This is because employees under strong organizational time norms usually cannot actively decide how to utilize their time. Otherwise, they might draw undesirable reactions from peers because of their time norm breaches [35].

With the help of a moderately strong temporal leadership, employees under strong organizational time norms are less likely to confront undesirable reactions from peers regarding their time utilization. This is because a moderately strong temporal leadership usually encompasses helping employees synchronize activities with their colleagues [29,43]. Additionally, this leads employees to attend to coordinating with each other while performing job duties [72]. In addition to synchronizing, a moderately strong temporal leadership also involves helping employees allocate temporal resources better [29,43]. In this way, employees can resolve conflicts with each other within the context of existing time constraints [29,38]. Therefore, we argue that employees under this context may less often feel pressed to always be ready for work during normal working hours. Following the second version of the spillover process [21,53], this friendly experience of work availability probably extends to the nonwork domain, such that employees are unlikely to suffer from dealing with work during off-time (i.e., EWA).

However, when coupled with an extremely strong temporal leadership, strong organizational time norms could aggravate the detrimental effect of temporal leadership on employee extended work availability. On the one hand, an extremely strong temporal leadership tends to closely monitor employees' work pace or supervise the time-use behaviors of employees. These task-oriented behaviors of a leader that strongly highlight temporality [29] may constrain employees' autonomous decisions on how to deal with temporal challenges at work. On the other hand, strong organizational time norms usually constraints employees' time-use behaviors through intense social pressure [35]. Within this context, employees cannot but follow the temporal requirements posed by organizations such to avoid undesirable reactions from peers [35]. Moreover, we expect that these two cases would jointly happen and further jeopardize employees' time-related job autonomy. Then, employees can be more inclined to experience intense pressure to always be ready for job demands because of the lack of temporal autonomy [35,73]. In line with the second version of spillover process [21,53], we argue that this experience would transfer into the nonwork domain, such that employees still need to suffer from dealing with work during off-time (i.e., EWA).

In sum, we expect that organizational time norms play a moderating role in the nonlinear association between temporal leadership and employees' EWA. For employees under strong organizational time norms, a moderately strong temporal leadership could be more inclined to alleviate employee EWA, while an extremely strong temporal leadership could be more inclined to intensify employee EWA. Hence, this leads to the following hypothesis:

**Hypothesis 5.** *Organizational time norms moderate the U-shaped association between temporal leadership and employee EWA, such that both the negative and positive associations are enhanced when organizational time norms are strong.*

## 3. Methodology

### 3.1. Participants and Procedures

To obtain a diverse sample, we used nonprobability purposive convenience sampling to recruit participants through the following procedure. First, we recruited seven senior students and one master student, who enrolled in methodology courses the first author teaches, from a university located in South China to perform as a survey team. They consented to help and would be awarded an extra grade point for the course they took. Second, we entrusted the survey team to recruit participants from their family members,

friends, and acquaintances. The candidates they recruited had to be white-collar, full-time workers, who might be contacted via a variety of ICTs to deal with work during off-time. If they had any doubt whether the potential invitee met our recruiting criteria, they needed to report to the first author and follow the decision made by the first author. Third, with the link that the survey team sent out, participants completed an online survey following the guidelines briefed by the survey team. Prior to responding to this survey, all invitees were informed that they were free to choose whether to participate and were able to quit the survey at any time. Moreover, invitees were also informed that their responses are anonymous, and the answers they provided would only be used for research purposes. At this stage, a total of 400 workers were invited and 259 responses were obtained. Drawing on DeSimone et al.'s recommendations [74], 7 responses in our initial sample were eliminated because of faster than one minute response times (about 2 s per item), 8 responses with ten or more invariant responses in a row were also dropped. Furthermore, given the sampling procedure used highly relies on the social networks of 9 students, another 4 responses with total working years and organizational tenure shorter than 3 months were eliminated because this usually indicates those invitees were probably interns.

Our final sample consisted of 240 full-time employees. This sample size is adequate given that it is larger than the required sample size of 160, which can be computed via G*power 3.1 package [75] using the "Linear multiple regression: Fixed model, $R^2$ deviation from zero" procedure according to a medium $f^2$ value of 0.15. Amongst these participants, 52.92% were male and 47.08% of them were female. The average age of our participants was 28.90 ($SD = 8.25$), and the average working years and organizational tenure was 6.70 years ($SD = 9.00$) and 5.14 years ($SD = 8.63$), respectively. In total, 48.75% of the participants had a bachelor's degree, 30.00% had an associate degree, 15.83% had only completed high school or lower-level education, and the remaining 5.42% had a postgraduate degree. Amongst these participants, 83.33% were nonmanager workers. The participants came from multiple types of organizations in mainland China, including private enterprise (58.33%), government and public institution (17.92%), state-owned enterprise (12.92%), and foreign-invested company (10.83%).

*3.2. Measurements*

All measures we used in the present study were originally developed in English. Then, we followed Brislin's procedures [76] to translate and back translate all measures into Chinese ones. To control for common method bias (CMB) in the current study, we followed Podsakoff et al.'s recommendations [77] and introduced psychological separation among the studied constructs by using somewhat different scale formats.

3.2.1. Employee EWA

Employee EWA was assessed with four items adopted from Dettmers et al.'s work [8] using a seven-point Likert scale ranging from 1 (Strongly disagree) to 7 (Strongly agree). An example item is "My supervisor expects me to be available for work related things also outside my regular working hours". The Cronbach's α coefficient for this scale is 0.91. Although the strong correlations of the third item (i.e., "Due to my job duties, I have to be available for work related things") with the second- ($r = 0.77$, $p < 0.001$) and fourth ($r = 0.76$, $p < 0.001$) item in the current study may undermine the fit of this one-factor construct to our survey data, most of the fit indices ($\chi^2$ (2) = 8.77, $p < 0.05$, CFI = 0.97, TLI = 0.93, RMSEA (90 percent CI) = 0.12 (0.06, 0.18), and SRMR = 0.03) are good enough, except for the RMSEA. Since this four-item scale for extended work availability has been validated by multiple studies [7–9], keeping all items in the current study should not be a serious problem.

3.2.2. Temporal Leadership

Temporal leadership was assessed with five items adapted from Mohammed and Nadkarni's seven-item team temporal leadership scale [29]. Rather than the team context in Mohammed and Nadkarni's original instrument [29], the items we used to measure tem-

poral leadership were modified to fit in the supervisor–subordinate context. An example item in our revised scale is "My immediate supervisor prioritizes tasks and allocate time to each task". All five items in our revised measure were rated using a six-point Likert scale ranging from 1 (Does not describe him/her at all) to 6 (Describes him/her very well). The Cronbach's $\alpha$ coefficient for this scale is 0.89. The fit indices ($\chi^2$ (5) = 6.18, $p$ = 0.29, comparative fit index (CFI) = 1.00, Tucker–Lewis index (TLI) = 1.00, root mean square error of approximation (RMSEA) (90 percent CI) = 0.03 (0.00, 0.09), and standardized root mean square residual (SRMR) = 0.02) well support the one-factor model of temporal leadership for the five items.

### 3.2.3. Employee Procrastination

Employee procrastination was measured with five items adapted from form G of Lay's procrastination instrument [78] using a six-point Likert scale ranging from 1 (Does not describe me at all) to 6 (Describes me very well). These items were derived from Renn et al.'s instrument for self-defeating behavior [79]. An example item is "In preparing for some deadlines, I often waste time doing other things". The Cronbach's $\alpha$ coefficient for this scale is 0.81. The fit indices ($\chi^2$ (4) = 8.09, $p$ = 0.09, CFI = 0.98, TLI = 0.96, RMSEA (90 percent CI) = 0.07 (0.00, 0.12), and SRMR = 0.04) well support the one-factor model of procrastination for the five items.

### 3.2.4. Organizational Time Norms

As organization time norms can be considered as an organization's time management environment [35], we used seven items derived from Burt et al.'s time management environment (TiME) scale [70] to measure individuals' perception of organizational time norms. The adapted scale in the current study encompasses two dimensions. One dimension captures the extent to which an organization supports the time management processes with three items, and the other captures the extent to which an organization emphasizes the value of time with four items. All seven items were rated using a seven-point Likert scale ranging from 1 (Strongly disagree) to 7 (Strongly agree). Example items for the adapted measure include "Making time to plan the days' work is encouraged in my organization" and "Emphasis is placed on keeping to deadlines". The Cronbach's $\alpha$ coefficient for this scale is 0.90. Additionally, the Cronbach's $\alpha$ coefficient for the subscales of the two dimensions (i.e., support for time management processes and time values) is 0.83 and 0.87, respectively. The fit indices ($\chi^2$ (13) = 24.74, $p$ < 0.05, CFI = 0.98, TLI = 0.97, RMSEA (90 percent CI) = 0.06 (0.03, 0.09), and SRMR = 0.03) well support the two-factor model of organizational time norms for the seven items.

### 3.2.5. Control Variables

Since certain individual characteristics and organizational contexts could induce employees to deal with work via ICT during off-time [18,80], participants' managerial role, perceived control of time, and abusive supervision were included for control. Specifically, the managerial role was measured as a dummy variable coded as 0 for nonmanager positions and 1 for manager positions. Perceived control of time refers to the extent to which an individual feels in control of time [81]. It was measured with four reverse-coded items adapted from Macan's work [81] using a six-point Likert scale ranging from 1 (never) to 6 (always). An example item is "I underestimate the time that it would take to accomplish tasks". The Cronbach's $\alpha$ coefficient for this scale is 0.81. The fit indices ($\chi^2$ (2) = 0.65, $p$ = 0.72, CFI = 1.00, TLI = 1.02, RMSEA (90 percent CI) = 0.00 (0.00, 0.06), and SRMR = 0.01) well support the one-factor model of perceived control of time for these four items. Abusive supervision denotes "subordinates' perceptions of the extent to which their supervisors engage in sustained display of hostile, verbal and non-verbal behaviors excluding physical contact (p. 178)" [82]. It was measured with Aryee et al.'s ten-item scale [83] ranging from 1 (never) to 5 (always). An example item is "My supervisor makes negative comments about me to others". The Cronbach's $\alpha$ coefficient for this scale is 0.94. The fit indices



($\chi^2$ (35) = 68.33, $p$ < 0.001, CFI = 0.97, TLI = 0.96, RMSEA (90 percent CI) = 0.06 (0.05, 0.08), and SRMR = 0.04) well support the one-factor model of abusive supervision for these ten items.

### 3.3. Analysis Strategies

We conducted all the analyses in the R environment (version 4.0.2 (http://www.r-project. org/ (accessed on 2 July 2021))) with RStudio. To assess the goodness of fit for both the scales and measurement model, we used the R package developed by Rosseel [84] for latent variable analysis (i.e., lavaan) to conduct confirmatory factor analysis (CFA). Given the small sample size in the present study, the following criteria for the fit indices were used: $\chi^2$/df < 5, both CFI and TLI $\geq$ 0.90, RMSEA $\leq$ 0.08, and the SRMR $\leq$ 0.06 [85,86]. Following Aiken and West's procedure [87], as well as relevant works on testing U-shaped relationships [88,89], we then performed a series of multiple hierarchical regressions to test the hypotheses.

## 4. Results

### 4.1. Confirmatory Factor Analysis of the Measurement Model

Prior to testing the hypotheses, we conducted a series of CFAs to examine the fit of the hypothesized measurement model with six latent constructs (i.e., employee EWA, temporal leadership, employee procrastination, organizational time norms, perceived control of time, and abusive supervision). The fit indices ($\chi^2$ (545) = 964.14, $p$ < 0.001, CFI = 0.91, TLI = 0.90, RMSEA (90 percent CI) = 0.06 (0.05, 0.06), and SRMR = 0.06) well supported the measurement model with items of these six constructs loaded only on their corresponding factor. Additionally, all standardized loadings of the items on their specified constructs were significant and ranged from 0.53–0.93. Moreover, the fit indices for our six-factor measurement model out-performed the one-, two-, three-, four-, and five-factor alternative models. These results lead us to conclude that the six constructs in our measurement model had a good fit to the data [85,86].

Given that all measures in the current study were obtained using a cross-sectional and self-reported design, we conducted a CMB analysis by adding an unmeasured method factor [77] on which all items loaded to our hypothesized six-factor measurement model. In comparing with the goodness of fit for our hypothesized measurement model, most of the fit indices ($\chi^2$ (544) = 968.45, $p$ < 0.001, CFI = 0.91, TLI = 0.90, RMSEA (90 percent CI) = 0.06 (0.05, 0.06), and SRMR = 0.06) improved slightly, except for $\chi^2$ ($\Delta\chi^2$ = 4.31, $\Delta df$ = 1, $p$ < 0.05), when the unmeasured method factor was added. However, the variance extracted by the unmeasured method factor was only 0.08, falling far below the commonly suggested 0.50 cutoff for the presence of a substantial common method bias [77]. Therefore, although we could not warrant that CMB did not present in our data, it would not be a serious problem when testing our hypotheses.

### 4.2. Descriptive Statistics and Correlations amongst the Study Variables

Table 1 presents the means, standard deviations, and correlations amongst the study variables. As expected, employee EWA is positively related to procrastination ($r$ = 0.40, $p$ < 0.001) and organizational time norms ($r$ = 0.30, $p$ < 0.001), respectively (see Table 1). These positive associations provide preliminary support for Hypothesis 2 and Hypothesis 4.

**Table 1.** Descriptive statistics and correlations among study variables.

| Variables | Mean | SD | 1 | 2 | 3 | 4 | 5 | 6 | 7 |
|---|---|---|---|---|---|---|---|---|---|---|
| 1. Managerial role | 0.17 | 0.37 | 1.00 | | | | | | |
| 2. Perceived control of time | 3.91 | 0.97 | 0.02 | 1.00 | | | | | |
| 3. Abusive supervision | 1.52 | 0.71 | −0.07 | −0.17 ** | 1.00 | | | | |
| 4. Employee EWA | 3.60 | 1.57 | 0.01 | −0.16 * | 0.16 * | 1.00 | | | |
| 5. Temporal leadership | 4.53 | 0.91 | 0.11 † | 0.15 * | −0.26 *** | 0.15 * | 1.00 | | |
| 6. Employee procrastination | 3.20 | 1.07 | −0.06 | −0.44 *** | 0.08 | −0.04 | 0.40 *** | 1.00 | |
| 7. Organizational time norms | 5.42 | 0.98 | 0.09 | 0.04 | −0.26 *** | 0.71 *** | 0.30 *** | 0.10 | 1.00 |

Notes: $n$ = 240. † $p$ < 0.10, * $p$ < 0.05, ** $p$ < 0.01, *** $p$ < 0.001.

### 4.3. Hypotheses Testing

Prior to the hypotheses testing, we computed the squared term of temporal leadership and its product term with procrastination or organizational time norms after mean-centering these variables. As can be seen in Table 2, after controlling for managerial role, perceived control of time, and abusive supervision, temporal leadership squared further accounts for thirteen percent of the variance of extended work availability ($\Delta R^2 = 0.13$, $p < 0.001$) in model 2. The coefficient on the squared term for temporal leadership is positive and significant ($\beta = 0.32$, $p < 0.001$), and the regression weight of temporal leadership is also positive and significant ($\beta = 0.32$, $p < 0.001$). While these results do not reflect a perfect U-shaped relationship between temporal leadership and employee EWA [87], the curvilinear association between temporal leadership and employee EWA can be further tested following Lind and Mehlum's three-step procedure [88] recommended by Haans et al. [89]. Calculation based on the standardized regression coefficients in Model 2 using the formula proposed by these works shows that all three conditions for a U-shaped relationship were held as follows. First, the squared term for temporal leadership has a positive and significant regression weight ($\beta = 0.32$, $p < 0.001$). Second, the slope of temporal leadership at the low end (−2.93) and high end (1.47) of the data range for temporal leadership after being mean-centered were significantly equal to −1.56 (i.e., $0.32 + 2 \times 0.32 \times (-2.93)$) and 1.26 (i.e., $0.32 + 2 \times 0.32 \times 1.47$), respectively. Third, the turning point of the curvilinear relationship between temporal leadership and employee EWA is at −0.50 (i.e., $-0.32 \div (2 \times 0.32)$), located well with the data range between −2.93 and 1.47. Therefore, our analyses provide support for Hypothesis 1, which stated that temporal leadership has a U-shaped association with employee EWA. This curvilinear relationship is shown in Figure 2.

**Table 2.** Regression results of the curvilinear relationship between temporal leadership and employee extended work availability.

| | Model 1 | Model 2 | Model 3 | Model 4 | Model 5 | Model 6 |
|---|---|---|---|---|---|---|
| Managerial role | 0.03 | 0.03 | 0.05 | 0.03 | 0.02 | 0.02 |
| Perceived control of time | −0.14 * | −0.16 ** | −0.01 | −0.05 | −0.13 * | −0.13 * |
| Abusive supervision | 0.14 * | 0.21 *** | 0.20 *** | 0.19 ** | 0.23 *** | 0.25 *** |
| Temporal leadership | | 0.32 *** | 0.29 *** | 0.26 *** | 0.06 | −0.05 |
| Temporal leadership squared | | 0.32 *** | 0.25 *** | 0.26 *** | 0.27 *** | 0.22 ** |
| Procrastination | | | 0.35 *** | 0.25 ** | | |
| Temporal leadership × Procrastination | | | | 0.25 *** | | |
| Temporal leadership squared × Procrastination | | | | 0.13 | | |
| Organizational time norms | | | | | 0.35 *** | 0.27 ** |
| Temporal leadership × Organizational time norms | | | | | | 0.12 |
| Temporal leadership-squared × Organizational time norms | | | | | | 0.24 * |
| *F* | 3.82 * | 10.45 *** | 14.96 *** | 13.68 *** | 12.09 *** | 9.80 *** |

Notes: $n = 240$. * $p < 0.05$, ** $p < 0.01$, *** $p < 0.001$. Standardized regression coefficients were reported.

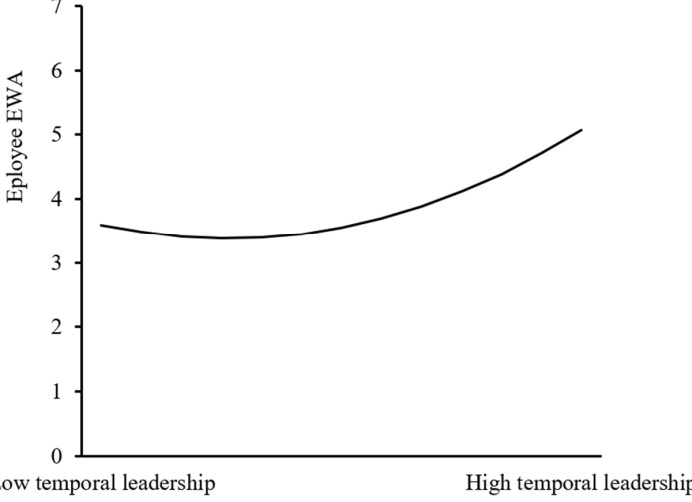

**Figure 2.** The curvilinear relationship between temporal leadership and employee EWA.

When testing the effect of employee procrastination, the positive coefficients on procrastination in model 3 ($\beta$ = 0.35, $p$ < 0.001) and model 4 ($\beta$ = 0.25, $p$ < 0.001) were both positive and significant, providing further support for Hypothesis 2, which states that procrastination has a positive association with employee EWA. However, the coefficient on the interaction term computed with temporal leadership squared and procrastination in model 4 is not significant ($\beta$ = 0.13, $p$ = 0.11). This does not support Hypothesis 3, which states that the U-shaped relationship between temporal leadership and employee EWA is moderated and intensified by employee procrastination.

When testing the effect of organizational time norms, the positive coefficients of organizational time norms in model 5 ($\beta$ = 0.35, $p$ < 0.001) and model 6 ($\beta$ = 0.27, $p$ < 0.01) were both positive and significant, providing further support for Hypothesis 4, which states that strong organizational time norms have a positive association with employee EWA. In addition, the inclusion of the interaction term computed with temporal leadership squared and organizational time norms in model 6 explains greater variance in employee EWA over and above model 5 ($\Delta R^2$ = 0.02, $p$ < 0.10). Furthermore, the interaction between temporal leadership squared and organizational time norms is positive and significant ($\beta$ = 0.24, $p$ < 0.05). These results provide support for Hypothesis 5, which states that the U-shaped relationship between temporal leadership and employee EWA is moderated and intensified by strong organizational time norms. Following Aiken and West [87], we probed the moderating effect by plotting the relationship between temporal leadership and employee EWA at one standard deviation above and below the mean of organizational time norms. As can be seen in Figure 3, when organizational time norms are strong, the U-shaped relationship between temporal leadership and employee EWA is attenuated.

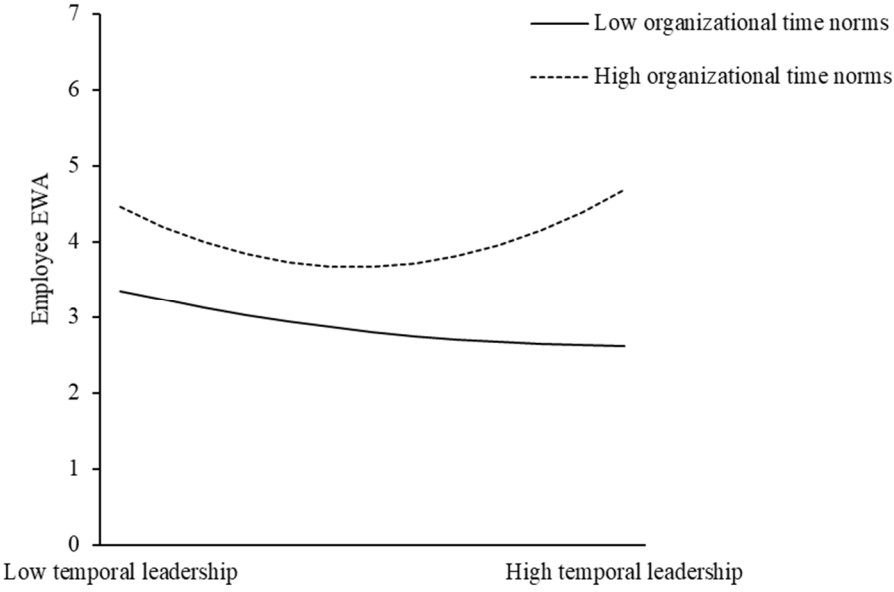

**Figure 3.** Interaction between temporal leadership and organizational time norms in predicting employee extended work availability.

To present the interaction effect of the squared term of temporal leadership and organizational time norms on employee EWA more precisely, we further probeb the conditional effect of the squared term of temporal leadership on employee EWA using the Johnson–Neyman technique [90]. As can be seen in Figure 4, when organizational time norms are strong but with values within their range of observed data (−3.00, 1.58) in the current study, i.e., greater than −0.30, the slope of the squared term of temporal leadership in Model 6 is positive significant ($p$ < 0.05). Even though the slope of the squared term of temporal leadership in Model 6 would be negative and significant ($p$ < 0.05) when organizational time norms' values are smaller than −16.00, this would not be probed in the

current study because the range of observed values of organizational time norms is $(-3.00, 1.58)$. In a sense, this result provides further support for the moderating role organizational time norms play in intensifying the positive association between temporal leadership and employee EWA when organizational time norms are strong.

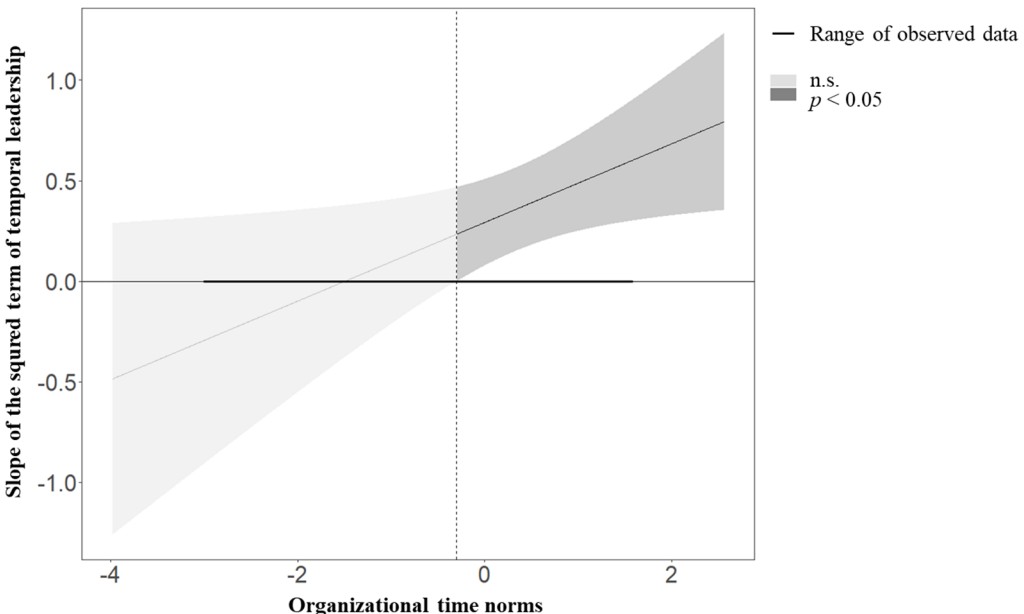

**Figure 4.** Conditional effect of the squared term of temporal leadership on employee EWA.

Moreover, supplementary analyses on the variance inflation factor (VIF) for each independent in our six listed regression models (see Table 2) show that no VIF value is greater than the common cut-off threshold of 10 [85]. Although we cannot warrant that there is no multicollinearity in our present study, it would not be a serious problem when deciding whether our hypotheses were supported by the analyses.

## 5. Discussion

Taking the temporal lens in organizational research [19], we focused on the roles three time-related constructs (i.e., temporal leadership, employee procrastination, and organizational time norms) play in determining employee EWA. With a sample of 240 full-time employees in mainland China, we confirmed the U-shaped association between temporal leadership and employee EWA. This finding not only provides empirical evidence for the possible integration of both positive and negative spillover processes stated by Lambert [40] into a same framework simultaneously, but also enriches our knowledge about the TMGT effect [30] of temporal leadership. Moreover, we also found both employee procrastination and organizational time norms are positively related to employee EWA. Additionally, we further found that strong organizational time norms strengthened the U-shaped association between temporal leadership and employee EWA. These findings shed light on the antecedents of employee EWA. However, the present study is only the first step to explore how managers can alleviate employee EWA in an ICT-prevalent work environment. Future studies should devote more efforts to exploring the inducing mechanisms of employee EWA by taking into consideration the relationship-oriented dimension of temporal leadership, as well as other leadership practices. Implications for theory and practice, as well as the limitation of this study are discussed in the following sections.

### 5.1. Theoretical Implications

The findings in our study have several theoretical contributions. First, the current study extends the literature regarding today's ICT-prevalent work environment and its effects by examining the time-related antecedents of employee EWA. Whereas previous

scholars have proposed certain organizational contexts and individual characteristics to shape employees' work-related ICT use beyond normal business hours [18,39], few have examined how these factors relate to employee EWA. Unlike prior research that focused on the consequences of employee EWA [6,7], our present study stresses the importance of the determinants of this phenomenon. Taking the temporal lens in organizational research [19], this study provides a promising theoretical framework for exploring how managers can mitigate employee EWA by helping and/or encouraging them to better utilize time resources while performing job duties.

Second, this study extends previous theoretical frameworks for explicating the spillover process linking work and nonwork domains. Although spillover theory posits that job conditions have both positive and negative direct effects on nonwork domain outcomes [40], few studies examine these two spillover processes simultaneously. To our knowledge, this study is the first one to integrate both positive and negative spillover effects in a same model by examining the U-shaped association between temporal leadership and employee extended availability. Focusing on the time-related experience spillover, the U-shaped association we found also resonates well with Edwards and Rothbard's argument about the resource drain for work and family time [21]. Taking into consideration the role of organizational time norms, we also found they are not only positively related to employee EWA but also intensify the foregoing U-shaped association when organizational time norms are strong. Since organizational time norms are the time management environment in organizations [35], these findings can complement the knowledge [21,40] regarding the effects of job conditions on nonwork domain outcomes to some extent.

Third, the present study contributes to the literature on temporal leadership by examining its curvilinear association with employee EWA. Although temporal leadership was identified as one of the promising areas of temporal organizational research [19], it seems to be somewhat under-discussed in comparison with other leadership practices (e.g., transformational leadership, ethical leadership, etc.). The U-shaped association between temporal leadership and employee extended availability we found extends Mohammed and Alipour's theoretical model on the effect of temporal leadership [66]. Although previous studies usually viewed temporal leadership as a positive leadership practice with desirable outcomes [29,33,41,42], temporal leadership is not always beneficial [91]. The curvilinear association between temporal leadership and employee EWA we found somewhat resonates with other similar curvilinear effects in recent temporal leadership research [92]. This finding also complements the extant literature by revealing its potential adverse effects. In line with the curvilinear effect of positive leader behavior on subordinate job satisfaction [93], we are further able to answer Pierce and Aguinis's call [30] for studying the TMGT effect in management.

*5.2. Practical Implications*

Our findings present several valuable implications for management practices, especially for improving management regarding the work–life balance of employees when the work–home boundary is increasingly blurred for white-collar workers. First, this study shows that temporal leadership had a U-shaped association with employee EWA. Thus, we suggest managers strive to pursue the Golden Mean [94,95] in terms of strength when displaying time-related leadership practices. A moderately strong temporal leadership should help in avoiding the TMGT effect [30] when alleviating employee EWA. In addition, managers should also take into consideration the socio-emotional needs of employees while managing their time top-down [66]. As such, managers can more easily to facilitate employees' effectiveness in meeting deadlines without pushing them too much to be available for work during off-time.

Second, results indicate that procrastination is positively related to employee extended work availability. Therefore, managers should find ways to help employees to overcome procrastination at work. For example, time management training programs are recommended to be introduced. By helping employees set and strive for time-related goals,

time management training has been found to enhance their time management capabilities [96]. As procrastination may be more rooted in behavioral styles, managers can further capitalize on cognitive behavioral therapy interventions to reduce procrastination in organization [97]. This approach involves helping employees explore how they feel about and behave towards procrastination and eventually change their procrastination behaviors into productive ones [98].

Third, we found strong organizational time norms not only induce employee extended work availability but also strengthen its U-shaped association with temporal leadership. While strong organizational time norms help in boosting organizational profitability [35,70], managers should pay attention to their potential downside of eliciting high-level employee time pressure. They can, for example, show empathy for the socio-emotional needs of employees while managing their time top-down [43]. They can also initiate constructive changes to organizational time norms in terms of taking into consideration employees' time-related experience at work. As such, managers can help counterbalance the adverse effects of organizational time norms in this context.

### 5.3. Limitations and Suggestions for Future Research

Our study has several limitations. First, focusing on the temporal lens in organizational research [19], we examined the direct and/or moderating effects of three time-related constructs (i.e., temporal leadership, employee procrastination, and organizational time norms) on employee EWA. Future research could incorporate other leadership practices (e.g., family-supportive supervisor behavior) as well as certain relevant constructs (e.g., workaholism and family-friendly policy) to investigate their impact mechanisms on employee extended work availability. In addition to the work-related determinants, taking into consideration certain aspects of daily life (e.g., living arrangements) will surely shed new light on this area. Additionally, this can also contribute to the field that investigates the relationship between individuals' time attitudes and their everyday life [69].

Second, we only included the task-oriented dimension of temporal leadership in the current study. Without the relationship-oriented dimension in the temporal leadership construct [29,43], we cannot easily conclude its TMGT effect [30] on employee extended work availability. To resolve this issue, future research could simultaneously take into consideration the effects of both the task-oriented and relationship-oriented dimensions of temporal leadership in this context.

Third, we conducted a survey on participants from the personal networks of a nine-person survey team using a cross-sectional design. Given the nonprobability of convenience sampling, the procedure we selected the participants with undermines the generalizability of our findings to other population groups. Randomized sampling is recommended for future research to overcome such a drawback. Moreover, the cross-sectional design we used cannot preclude the probability of the opposite direction of effects regarding the hypothesized associations. Future research could use a longitudinal approach to collect multiwave data and further examine the causal relationship with more sophisticated analytical strategies (e.g., cross-lagged panel model).

**Author Contributions:** The data analysis and manuscript were prepared by J.Z. and S.Z. All authors listed have made substantial and intellectual contributions to the work and approved the final version. All authors have read and agreed to the published version of the manuscript.

**Funding:** This research was funded by the 13th five-year Planning Project for the Development of Philosophy and Social Sciences of Guangdong Province (NO. GD19CGL20), and the 13th five-year Planning Project for the Development of Philosophy and Social Sciences of Guangzhou municipal (NO. 2019GZYB87).

**Institutional Review Board Statement:** Not applicable.

**Informed Consent Statement:** Not applicable.

**Data Availability Statement:** Data that support the findings of this study are available from the corresponding author upon reasonable request.

**Acknowledgments:** The authors express thanks to the survey team for their great help in recruiting participants.

**Conflicts of Interest:** The authors declare that the research was conducted in the absence of any commercial or financial relationships that could be construed as a potential conflict of interest.

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
