# Peer review of "Make Time for Employees to Be Sustainable: The Roles of Temporal Leadership, Employee Procrastination, and Organizational Time Norms"

_sustainability, doi:10.3390/su14148778_

Round 1

Reviewer 1 Report

1. Abstract must be more informative in the sense of research question as well as authors must provide data regarding their direct final application.

2. Please strengthen INTRODUCTION section, especially use the literature from the cognate discipline to build your arguments. The author(s) missed some most relevant studies. Here are some of the literature from the cognate discipline that will help strengthen the arguments as: 

Kühnel, J., Bledow, R., & Kuonath, A. (2022). Overcoming Procrastination: Time Pressure and Positive Affect as Compensatory Routes to Action. Journal of Business and Psychology, 1-17.

Eniola, O. E. (2022). Employee Engagement in the Home-Work Lifeworld. International Business Research15(6), 1-49.

Song, Z., Nida, G., Asghar, M., Sarfraz, M., & Rafique, M. A. (2022). Polychronicity, Time Perspective, and Procrastination Behavior at the Workplace: An Empirical Study. Anales de Psicología/Annals of Psychology38(2), 355-364.

3-You'd better discuss the theoretical gap in the introduction section. Second, discuss the potential contributions made to the literature before the end. More specifically, how this study is contributing to the existing literature? 

4. The authors should add a table to provide the data information for better readability.

5. Within the text start sentences with authors' names, always give a logical flow between sentences

6.better discuss concepts such as sustainable employees, Employee Procrastination

7. conclusions report the limitations of your work

Author Response

Point 1: Abstract must be more informative in the sense of research question as well as authors must provide data regarding their direct final application.

Response 1: Thanks for your insightful suggestion!

We have revised the abstract by including our research question: how managers can alleviate employee extended work availability (EWA) in the ICT prevalent work environment? Unfortunately, we are afraid the data regarding their direct final application is not applicable for the present study. But we did report some critical coefficients regarding the associations amongst the research variables in the revised abstract. We hope this revision has made the abstract more informative.

Thanks again for your constructive comment on this issue.

Point 2: Please strengthen INTRODUCTION section, especially use the literature from the cognate discipline to build your arguments. The author(s) missed some most relevant studies. Here are some of the literature from the cognate discipline that will help strengthen the arguments as: 

Kühnel, J., Bledow, R., & Kuonath, A. (2022). Overcoming Procrastination: Time Pressure and Positive Affect as Compensatory Routes to Action. Journal of Business and Psychology, 1-17.

Eniola, O. E. (2022). Employee Engagement in the Home-Work Lifeworld. International Business Research15(6), 1-49.
Song, Z., Nida, G., Asghar, M., Sarfraz, M., & Rafique, M. A. (2022). Polychronicity, Time Perspective, and Procrastination Behavior at the Workplace: An Empirical Study. Anales de Psicología/Annals of Psychology38(2), 355-364.

Response 2: Thanks for your constructive suggestion and the recommended reference!

Drowing on Eniola’s (2022) work, we have rewritten the argument on the connection between time and employee EWA in paragraph 4 (lines 86-87) within the INTRODUCTION section.

Drowing on Kühnel et al.’s (2022) work, we have rewritten the argument on the role of procrastination in our conceptual model to a great extent. Specifically, sentences have been added to propose the association between procrastination and employee EWA. And the arguments on the moderating role of procrastination plays in the temporal leadership-employee EWA linkage have been rewritten by taking into account different levels of positive affect induced by moderate and extremly strong temporal leadership, respectively. We hope this revision would clarify the role of procrastination plays in determing employee EWA better.

Drowing on Song et al.’s (2022) work, we have added argument on the association between organizational time norm between employee EWA by taking the positive polychronicity-procrastinating behavior linkage as an example support. We hope this revision would clarify the role of organizational time norm plays in determing employee EWA better.

Thanks again for your constructive comment and recommended reference on this issue. We hope the INTRODUCTION section has been strengthened to some extent.

Point 3: You'd better discuss the theoretical gap in the introduction section. Second, discuss the potential contributions made to the literature before the end. More specifically, how this study is contributing to the existing literature?

Response 3: Thanks for your constructive suggestion!

Although we have discussed the theoretical gap and the potential contributions in the INTRODUCTION section, we need to admit the weakness without well taking into account the relavant research that you recommended.

In the revised manuscript, we have rewritten the theoretical gaps (Paragraph 3 in the INTRODUCTION section) to some extent by refering to Eniola’s (2022) work. By strengthing the essential role time plays in understanding how employees engage in home-based work as well as its consequences (Eniola, 2022), we hope the theoretical gap regarding the time-related determinants of employee EWA has been justified better.

As for the potential contributions (Paragraph 6 in the INTRODUCTION section) we made to the literature, we start with a relative “big picture” by stating “it proposes a promising framework for exploring the determinants of employee EWA by taking the temporal lens in organizational research”. Focusing on the relevant field of work-home interaction, we then discussed the potential contribution to extend the theoretical framework of spillover theory. Because we integrate both positive and negative spillover effects into a same model simultaneously. At the end of this part, we mainly discuss the potential advancement we made to the emerging temporal theory through providing some interesting evidence on its effects.

All these three potential contributions could be made around answering our research question: How managers can alleviate employee EWA in the ICT prevalent work environment? Details about the theorization of our conceptul model can be refer to the 4th and 5th paragraph in the INTRODUCTION section.

By taking into account your constructive suggestion, we have rewritten multiple arguments based on the useful references you recommended. We hope this revised manuscript would make our discussion on the theoretical gap and potential contribution to the literature much clearer.

Thanks again for your constructive comment on this issue. It will definitely help us do better research in the future.

Point 4: The authors should add a table to provide the data information for better readability.

Response 4: Thanks for your constructive suggestion!

But we are afraid that more tables might distract readers’ attention from the critical results we found in the current study. And we need to admit that we are not sure which table should be added as you suggestion. If they are necessary, we would be absolutely happy to provide the relevant information in other tables.

Thanks again for your suggestion on this issue.

Point 5: Within the text start sentences with authors' names, always give a logical flow between sentences.

Response 5: Thanks for your constructive suggestion!

We have read through the manuscript multiple times and made necessary changes accordingly. We hope the revised manuscript would be able to give a logical flow regarding this issue.

Thanks again for your constructive comment on this issue.

Point 6: better discuss concepts such as sustainable employees, Employee Procrastination.

Response 6: Thanks for your insightful comment!

We need to admit the weak discussion on some important concepts in the previous version of our manuscript. In this revised manuscript, we cited two references and added a statement in lines 51-52: “which might further inhibits employees from sustainably making contributions to the organization with a high level of thriving at work (vitality and learning)”. As we did not intend to propose a formal, scientific definition of sustainable employee, the aforementioned statement helps clarify this issue to some extent.

As for employee procrastination, we mainly use Steel’s (2007) definition. Taking into account the references you recommended, we have strengthened the arguments on the connection between employee procrastination and employee EWA. We hope this revision gives better discussions on employee procrastination in our manuscript.

Thanks again for your constructive suggestions on this issue.

Point 7: conclusions report the limitations of your work.

Response 7: Thanks for your insightful comment!

We need to admit that the conclusion part in the previous version of manuscript might be confusing. That is because we discussed the limitaions of our work at the end of the DISCUSSION section. Then we used a short CONCLUSION section to close the main body of our manuscript. To make the discussion clearer, we delted the last section named “Conclusion” in the previous version of our manuscript. In addition, we made revision to the DISCUSSION section where is needed to some extent. We hope the revised manucript would not confuse you on this issue.

Thanks again for your constructive comment on this issue. It will definitely help us do better research in the future.

Reviewer 2 Report

Thank you very much for the article for review. I read it with interest

Congratulations to the authors of the idea.

The article  Make Time for Employees to be Sustainable: The Roles of Temporal Leadership, Employee Procrastination, and Organizational Time Norm” deals with a very important and interesting topic, both in science and in management practice.

 Please try adding more recent literature on leadership.

line 87:   Before "temporal leadership" is defined, please explain to readers:  - what is workplace leadership,  - what are the types of leadership,  - why different leadership styles are changing  - what are the contemporary trends - - how can we place "temporary leadership" in the leadership theory 

Author Response

Point 1: Thank you very much for the article for review. I read it with interest

Congratulations to the authors of the idea.

The article  “Make Time for Employees to be Sustainable: The Roles of Temporal Leadership, Employee Procrastination, and Organizational Time Norm” deals with a very important and interesting topic, both in science and in management practice.

Response 1: Thanks for your nice comment and encouragement!

Point 2:  Please try adding more recent literature on leadership.

Response 2: Thanks for your insightful suggestion!

Taking in to your suggestion, we mainly include recent literature review on the leadership theory in the INTRODUCTION section. We hope this would help readers understand the field of leadership research and its contemporary trends, as well as the position of temporal leadership in the leadership theory. In addition, we cited some recent works on temporal leadership when discussing the theoretical implications in the DISCUSSION section. We hope this would better explain the connection between our findings with recent relevant works.

Thanks again for your constructive suggestions on this issue.

Point 3:  line 87: Before "temporal leadership" is defined, please explain to readers:  - what is workplace leadership,  - what are the types of leadership,  - why different leadership styles are changing  - what are the contemporary trends - - how can we place "temporary leadership" in the leadership theory .

Response 3: Thanks for your insightful suggestion!

We need to admit the weak discussion on the position of temporal leadership in the leadership theory. In this revised manuscript, we firstly brief the broad concept of workplace leadership in Line 92-94 by stating “The discussion of workplace leadership usually includes a variety of influence process that occurs in organizations to warrant the achievement of common goals”. Then, we cited literature review papers on leadership to explain behavioral component in leadership has been increasingly emphasized in the recently emerging leadership theories (Lines 94-98). After that, we introduced the definition of temporal leadership used in our manuscript (Lines 98-99). We hope this revision gives better discussions on temporal leadership and its connection with the leadership theories, and the contemporary trends in the field of leadership research.

Thanks again for your constructive suggestions on this issue. It will definitely help us do better research in the future.

Reviewer 3 Report

Dear author(s),

Thank you for the manuscript. It is interesting. However, few improvements would further help in the overall quality.

First, how do you check the credibility, validity, and authenticity of selected 240 full-time white collar employees? How did you ensure transparency in the selection process and how you avoid selection biases?

You did not explain the sampling technique and justification of sample size being adequate to draw logical conclusion? 

The managerial implications should not be in the discussion section. Prolong discussion section and critically evaluated present findings in the light of existing literature.

The limitations should come after conclusion.

There should be a dedicated section to inform the readers how the research field, concern industry, and overall economy of the world benefits from it. Theoretical and practical implications should be given after conclusion section. 

Good luck

Author Response

Point 1: Dear author(s),

Thank you for the manuscript. It is interesting. However, few improvements would further help in the overall quality.

Response 1: Thanks for your nice comment on our manucript!

Point 2:  First, how do you check the credibility, validity, and authenticity of selected 240 full-time white collar employees? How did you ensure transparency in the selection process and how you avoid selection biases?

Response 2: Thanks for your instructful comment!

We need to admit issues regarding the procedure that we used to recruit participants might exist. In fact, we did take measures to ensure the effectiveness of our selection process. Specifically, the first author kept in touch with each member of the survey team closely. He supervised the recruiting process and help the team decide whether their potential invitees meet our recruiting criteria, that said white collar, full-time workers. More details about the sampling procedure have been added to the revised manuscript. We hope this would somewhat help ensure transparency in our selection process. At this point in time, however, we are afraid that we cannot easily solve this problem. But we have taken into account this issue and discussed it in the limitation part.

Thanks again for your insightful comment on this issue. It will definitely help us do better research in the future.

Point 3:  You did not explain the sampling technique and justification of sample size being adequate to draw logical conclusion? 

Response 3: Thanks for your constructive comments!

Taking into account your comment, we briefed the sampling technique used in the current study by stating “we use non-probability purposive convenience sampling to recruit participants“ in the revised manuscript. This technique should be able to obtain a diverse sample.

In order to justify the adequacy of our sample size, we employ G*Power 3.1 package to compute the required sample size for testing our regression model with up to 8 predictors. The result shows that our sample size of 240 is larger than the required sample size of 160 according to a medium effect. Discussion on this issue has beed added to our revised manuscript.

Thanks again for your comment on this issue. We hope our revisions give a better explanation about it.

Point 4:  The managerial implications should not be in the discussion section. Prolong discussion section and critically evaluated present findings in the light of existing literature. 

Response 4: Thanks for your constructive suggestion!

We need to admit it is somewhat weak in the discussion section. Taking into account your suggestion, we rewrit this section to some extent. After briefing the main points of the current study using a small paragraph at the beginning of the discussion section, we then sequently discuss three topics as follows, theoretical implications, practical implication, and limitations and suggestions for future research. In the revised manuscript, we strengthened the theoretical implication to some extent by including another two recent research on temporal leadership. But we are afraid the limitation part should be kept in the whole discussion section and used to close this section.

Thanks again for your insightful comment on this issue. It will definitely help us do better research in the future.

Point 5:  The limitations should come after conclusion. 

Response 5: Thanks for your insightful suggestion!

We need to admit that the conclusion part in the previous version of manuscript might be confusing. That is because we discussed the limitaions of our work at the end of the DISCUSSION section. Then we used a short CONCLUSION section to close the main body of our manuscript. To make the discussion clearer, we delted the last section named “Conclusion” in the previous version of our manuscript. In addition, we made revision to the DISCUSSION section where is needed to some extent. We hope the revised manucript would not confuse you on this issue.

Thanks again for your constructive comment on this issue. It will definitely help us do better research in the future.

Point 6:  There should be a dedicated section to inform the readers how the research field, concern industry, and overall economy of the world benefits from it. Theoretical and practical implications should be given after conclusion section. 

Response 6: Thanks for your insightful suggestion!

Taking into account your suggestion, we made somewhat revision to the last part of our previous version of manuscript. We hope some issues has been explained in the other responses. If there is still anything needs to be clarified, we would absolutely happy to made further revision on this issue.

Thanks again for your constructive comment on this issue. It will definitely help us do better research in the future.

Reviewer 4 Report

This article was a meaningful research topic, and overall it was well-formed. I hope that the following comments will help to revise it.

Compared to the correlation coefficients in Table 1, it seems that the regression coefficients in Table 2 are mathematically too large. In general, I recommend reviewing the standardized regression coefficient considering that it is not larger than the correlation coefficient.

It is thought that the quality of this paper will be higher if the figure 2 etc. are presented precisely using the Johnson and Neyman's floodlight. Moreover, It is recommended that your curvlinear relationships be explained in detail by referring to the following studies:

Haans, R. F. J., Pieters, C. and He, Z. L. (2016), "Thinking about U: Theorizing and testing U- and inverted U-shaped relationships in strategy research", Strategic Management Journal, Vol. 37 No. 7, pp. 1177-1195. 

Author Response

Point 1: This article was a meaningful research topic, and overall it was well-formed. I hope that the following comments will help to revise it.

Response 1: Thanks for your nice comment and encouragement!

Point 2:  Compared to the correlation coefficients in Table 1, it seems that the regression coefficients in Table 2 are mathematically too large. In general, I recommend reviewing the standardized regression coefficient considering that it is not larger than the correlation coefficient.

Response 2: Thanks for your insightful comment!

We also found that the standardized regression coefficients in Table are larger than the correlation coefficients in Table 1. But given the fact that both squared term and product term were included in most of these regression model, the potential multicollinearity could be an issue. However, we conducted supplementary analyses on the variance inflation factor (VIF) for each independent in our six listed regression model (see Table 2). Results found that no VIF value is greater than the common cut-off threshold of 10. Drawing on Hair et al.’ (2018) work, it would not be a serious problem when deciding whether our hypotheses were supported by the analyses.

In the revised manuscript, we have added relevant details to the end of Results section. At this point in time, we hope this revision by taking into account your concern could give better explanation about this issue.

Thanks again for your constructive comment on this issue. It will definitely help us do better research in the future.

Point 3:  It is thought that the quality of this paper will be higher if the figure 2 etc. are presented precisely using the Johnson and Neyman's floodlight. Moreover, It is recommended that your curvlinear relationships be explained in detail by referring to the following studies:

Haans, R. F. J., Pieters, C. and He, Z. L. (2016), "Thinking about U: Theorizing and testing U- and inverted U-shaped relationships in strategy research", Strategic Management Journal, Vol. 37 No. 7, pp. 1177-1195. 

Response 3: Thanks for your insightful suggestion!

By taking into account your suggestion and the recommed reference, we have added analyses using the Johnson-Neyman technique to probe the conditional effect of the squared term of temporal leadership on employee EWA. We hope this could better present the moderating role organizational time norm plays in intensifying the curvilinear association between temporal leadership and employee EWA.

Thanks again for your constructive comment on this issue.

Round 2

Reviewer 1 Report

No further comments